# Regulated Necrotic Cell Death in Alternative Tumor Therapeutic Strategies

**DOI:** 10.3390/cells9122709

**Published:** 2020-12-17

**Authors:** Yunseo Woo, Hyo-Ji Lee, Young Mee Jung, Yu-Jin Jung

**Affiliations:** 1Department of Biological Sciences, Kangwon National University, Chuncheon 24341, Korea; yunseo@kangwon.ac.kr (Y.W.); koko7912@kangwon.ac.kr (H.-J.L.); 2Kangwon Radiation Convergence Research Support Center, Kangwon National University, Chuncheon 24341, Korea; ymjung@kangwon.ac.kr; 3Department of Chemistry, Kangwon National University, Chuncheon 24341, Korea; 4BIT Medical Convergence Graduate Program, Kangwon National University, Chunchon 24341, Korea

**Keywords:** apoptosis, autophagy, necrosis, necroptosis, pyroptosis, ferroptosis, therapy-resistant tumors

## Abstract

The treatment of tumors requires the induction of cell death. Radiotherapy, chemotherapy, and immunotherapy are administered to kill cancer cells; however, some cancer cells are resistant to these therapies. Therefore, effective treatments require various strategies for the induction of cell death. Regulated cell death (RCD) is systematically controlled by intracellular signaling proteins. Apoptosis and autophagy are types of RCD that are morphologically different from necrosis, while necroptosis, pyroptosis, and ferroptosis are morphologically similar to necrosis. Unlike necrosis, regulated necrotic cell death (RNCD) is caused by disruption of the plasma membrane under the control of specific proteins and induces tissue inflammation. Various types of RNCD, such as necroptosis, pyroptosis, and ferroptosis, have been used as therapeutic strategies against various tumor types. In this review, the mechanisms of necroptosis, pyroptosis, and ferroptosis are described in detail, and a potential effective treatment strategy to increase the anticancer effects on apoptosis- or autophagy-resistant tumor types through the induction of RNCD is suggested.

## 1. Introduction

Cancer is one of the major causes of death worldwide, and it leads to approximately 1 in 6 deaths globally [1]. In 2018, 9.6 million people died from various types of cancer such as lung (1.76 million), colorectal (862,000), stomach (783,000), liver (782,000), and breast (627,000) cancer [2]. In general, cancer is induced by not only genetic factors but also physical, chemical, and biological carcinogens including radiation, asbestos, and viruses [3]. However, if these harmful factors are avoided and early screening for cancer prevention is performed thoroughly, 30~50% of tumors can be prevented, thus reducing the cancer burden [4]. Cancer patients are treated with surgery, chemotherapy, immunotherapy, and radiotherapy to relieve pain or symptoms [5]. The ultimate goal of these therapies is to kill cancer cells in the patients [6]. However, it is difficult to completely kill all cancer cells, since some cells are resistant to cancer therapies through various survival mechanisms [7].

Various types of cell death can be induced (Figure 1). Traditionally, the goal of anticancer therapy has been to induce apoptosis in cancer cells; however, cancer cells can avoid this type of death by inhibiting the activation of apoptotic signals [8]. In various types of cancer cells, the basal expression level of anti-apoptotic proteins such as survivin, PED, Bcl-xL, Bcl-2, and cFLIP, is high, and the expression of pro-apoptotic proteins such as bax, bim, bak, bad, puma, and noxa is reduced [9]. IL-4 and IL-6 derived from cancer cells can also function as anti-apoptotic factors. IL-4 production by primary epithelial cancer cells obtained from human colon, lung, and breast tissues elevates the expressions of anti-apoptotic proteins and increases the therapeutic resistance to anticancer agents, such as oxaliplatin, doxorubicin, and etoposide, in vitro and in vivo [10]. IL-6 secretion by ovarian, prostate, and lung cancer cells makes them resistant to cisplatin in vitro [11]. Moreover, therapy-induced apoptosis of cancer cells may promote tumor growth. Apoptotic extracellular vesicles (ApoEVs) released from glioblastoma (GBM) cells after chemotherapy or radiotherapy can cause the transformation of adjacent cells into a therapy-resistant population via alteration in the splicing process in mice [12]. Sphingosine-1 phosphate (S1P), which is secreted by cancer cells during apoptosis, was shown to stimulate S1P1 on primary human macrophages and to activate the angiogenic program [13]. Therefore, alternative strategies are needed to treat patients with apoptosis-resistant cancer.

Autophagy is an intracellular catabolic process that responds to various cellular stresses [14]. In cancer therapy, autophagy is involved in the survival and death of cancer cells, and the activation of a certain cell death type may be beneficial or detrimental to the cancer cells depending on the properties of the cancer microenvironment [15]. The induction of autophagy has been used as an alternative strategy to treat apoptosis-resistant cancer cells. TLRagonist-induced autophagy in tumors enhanced the efficiency of radiotherapy and improved anticancer immunity in vitro and in vivo [16,17]. Alkaloid compounds caused cell death in apoptosis-defective human cancer cell lines through the activation of autophagy [18]. Furthermore, the inhibition of autophagy may induce apoptosis in cancer cells. Chloroquine, an autophagy flux inhibitor, increased cyclophosphamide-induced apoptosis in mouse lymphoma cells lacking p53 activity and enhanced the radiosensitivity of human urinary bladder cancer cell lines, EJ and T24, via apoptosis [19,20]. Additionally, 3-methyladenine (3-MA), an autophagy inhibitor, promoted cisplatin-induced apoptosis in esophageal squamous cell carcinoma cells [21]. However, in contrast to these results, the inhibition of autophagy may make cancer cells more susceptible to cancer therapies. 3-MA significantly reduced the survival of radiation-exposed MDA-MB-231 cells, a human breast cancer cell line, and vincristine-treated leukemia cells [22,23]. These data suggest that the activation of autophagic mechanisms can result in various consequences for cellular fate. In addition to regulated non-necrotic cell death including apoptosis and autophagy, the induction of regulated necrotic cell death (RNCD) has been evaluated in recent studies to improve cancer therapy. The types of RNCD and corresponding therapeutic strategies will be discussed in this review.

## 2. Types of Regulated Necrotic Cell Death

Cell death is one of the major homeostatic mechanisms, and the appropriate regulation of cell death in cancer therapy is paramount [24]. Since accidental cell death (ACD) such as necrosis can promote the development and progression of cancer, the induction of regulated cell death (RCD) may be more effective for cancer therapy [25,26]. Unlike ACD, RCD is a cell death type controlled by specific proteins [27]. For decades, researchers in the field of anticancer therapeutics mainly studied apoptotic and autophagic types of cell death. However, resistance to apoptosis has been detected in various types of cancers during radiotherapy and chemotherapy [28]. New alternative therapeutic strategies are required, because the basal level of autophagy is dependent on the cancer type and can result in completely different treatment results [29]. RNCD is controlled by specific proteins similar to apoptosis or autophagy; however, the morphological features of RNCD are different, because it causes plasma membrane disruption [30]. Another significant difference is that RNCD induces higher inflammatory responses compared to those induced by apoptosis or autophagy [31]. 

Recent studies demonstrated that inflammatory signals induced by RNCD delay tumor progression and promote anticancer immune responses in mice and human cancer cells [32,33,34]. These results indicate that the induction of RNCD, including necroptosis, pyroptosis, and ferroptosis, may be a useful strategy of cancer therapy. Necroptosis, pyroptosis, and ferroptosis are commonly regulated by specific proteins and are accompanied by disruption of the cellular membrane (Figure 2). 

### 2.1. Necroptosis

Necroptosis is a type of programmed necrosis and its morphological features include cell death induced by cell expansion and plasma membrane rupture [35]. However, unlike necrosis, cell membrane permeability is tightly controlled during necroptotic cell death. Necroptosis can be induced by the TNF receptor superfamily, TLRs, interferon receptors, etc. [36]. In general, stimulatory factors, including ligands of these receptor or various stressors, can trigger extrinsic apoptosis through the activation of caspase-8 [37,38,39]. However, caspase-8 cleaves and thus inhibits the function of RIPK1 and RIPK3 proteins that are essential for the execution of necroptosis [40]. Therefore, the initiation of necroptosis can depend on the expression and activity of caspase-8 [41]. In most cancer cells, the expression or activity of caspase-8 can be suppressed or turned off [42]. Thus, cancer cells can be resistant to anticancer drugs that cause apoptosis [43]. When caspase-8 is absent or inactivated, RIPK1 and RIPK3 are phosphorylated and subsequently form a necrosome [44]. Activated RIPK3 phosphorylates the serine/threonine residues in MLKL to induce MLKL oligomerization [45]. The structural stability of the MLKL polymer depends on the disulfide bond between the molecules [46]. However, disulfide bonds are important for tetramerization of MLKL, but not for octamerization, which is important for membrane disruption [47]. Several mechanistic models of membrane collapse induced by MLKL have been proposed; however, specific consensus has not been reached [48]. After the phosphorylation of MLKL, the N-terminal domain is apparently translocated into the cell membrane [49]. MLKL pores have been confirmed to act as cation channels in liposomes [50]. In particular, the MLKL channel was shown to have higher permeability to Mg^2+^ than that to Ca^2+^ in the presence of Na^+^ and K^+^ [51]. Thus, insertion of an MLKL oligomer into the cell membrane can initiate necroptosis and subsequent release of damage-associated molecular patterns (DAMPs) containing intracellular components from the cells.

### 2.2. Pyroptosis

Both apoptosis and pyroptosis are mediated by caspases; however, unlike apoptosis, pyroptosis involves inflammatory caspase-related cell death [52]. Pyroptosis is distinguished from necroptosis, another RNCD, by involvement of caspases in cell death events. Unlike necroptis, pyroptosis involves plasma membrane disruption mediated by gasdermin (GSDM) proteins [53]. In general, pyroptosis includes inflammasome activation and maturation of IL-1β and IL-18 [54]. The assembly of the inflammasome complex is triggered by sensing cytoplasmic pathogen-associated molecular patterns (PAMPs) or DAMPs [55]. The NLR family proteins, such as NLRP1, NLRP3, and NLRC4, are responsible for the detection of these patterns [56]. Activated NLRs are oligomerized through the interaction between their NACHT domains, and then the CARD domain interacts with the CARD domain of pro-caspase-1 [57] to recruit pro-caspase-1 into the NLR oligomer [58]. Some NLRs have the PYD domain instead of the CARD domain [59]. These NLRs require ASC, an adaptor protein with the CARD domain, to recruit pro-caspase-1 [60]. In this case, the CARD domain of ASC interacts with the CARD domain in pro-caspase-1 [61]. Pro-caspase-1 in the inflammasome complex can be cleaved into two p20 and two p10 molecules, and the cleaved molecules bind to each other and act as a heterodimeric enzyme [62]. Activated caspase-1 cleaves IL-1β, IL-18, and GSDMD, an executor of pyroptotic cell death [63]. Furthermore, in humans, GSDMD can be cleaved by caspase-4, -5, and -11 [64]. The N-terminal domain of cleaved GSDMD is translocated into the plasma membrane and then oligomerized [65]. Finally, gasdermin pores mediated the K^+^ efflux and release of DAMPs [66]. In addition to GSDMD, GSDME can mediate pyroptosis without the formation of inflammasome complexes [67].

### 2.3. Ferroptosis

Ferroptosis is a type of RNCD caused by disruption of the plasma membrane [68]. However, unlike necroptosis and pyroptosis, ferroptotic cell death is not mediated by specific protein-related pores. Ferroptosis is caused by membrane disruption due to the accumulation of iron-dependent lipid peroxides [69]. Ferroptotic cell death depends on iron levels in the cells and can also be suppressed by iron chelators [70]. Extracellular Fe^3+^ atoms are bound to transferrin, a monomeric glycoprotein, and Fe^3+^ can be transferred into the endosome via the transferrin receptor (TfR) on the plasma membrane [71]. In the endosomes, Fe^3+^ is reduced to Fe^2+^ by endosomal ferrireductase [72]. Endosomal Fe^2+^ is released into the cytoplasm via a divalent metal transporter, SLC11A2 [73]. Cytosolic Fe^2+^ is oxidized to Fe^3+^ in the reaction with H_2_O_2_, resulting in the formation of •OH and HO^-^ in a process known as the Fenton reaction [74]. These radicals generated by the chemical reaction disrupt the plasma membrane through the peroxidation of membrane lipids [75]. Ferroptosis is closely associated with imbalance in the iron metabolism; however, it can also be caused by NADPH-mediated lipid peroxidation [76]. Moreover, the inhibition of glutathione, a homeostatic molecule scavenging intracellular reactive oxygen species (ROS), can induce ferroptosis [77]. Overall, peroxidation of membrane lipids due to excessive ROS generation can cause ferroptosis.

## 3. Regulated Necrotic Cell Death and Tumor Therapy

Applications of the strategies for induction of apoptotic and autophagic cell deaths have been investigated in the treatment of various cancer types. However, since cancer can be resistant to apoptosis, new therapeutic approaches should be investigated. In particular, unlike apoptosis and autophagy, RNCD includes cell death types that induce inflammatory responses; hence, it is necessary to consider whether RNCDs are suitable for cancer treatment.

### 3.1. Necroptosis in Tumor Therapy

Necroptosis has been reported to induce the inhibition of tumor growth in various cancer types. Ionizing radiation caused necroptosis in human thyroid and adrenocortical cancers [78]. Moreover, radiation regulated the repopulation of human colorectal cancer cells via the RIP1/RIP3/MLKL/JNK/IL-8 pathway [79]. Chemoresistance of cancer cells is strongly associated with high levels of enzymatic activity of aldehyde dehydrogenase (ALDH) [80]. ALDH1A family inhibitors (ALDH1Ai), such as DEAB, 673A, and CM10, increased the sensitivity of chemotherapy-resistant human ovarian cancer cells, SKOv3, to cisplatin through the induction of calcium-dependent necroptosis [81]. In particular, a Smac mimetic necroptosis inducer increased the sensitivity of apoptosis-resistant cells, including RIPK3-expressing KKU213, RMCCA-1, and HuCCT-1 cell lines, to TNF in vitro [82]. Staurosporine, shikonin, proteasome inhibitors (MG132 and bortezomib), and Smac mimetics suppressed the viability of leukemia cells by inducing necroptosis in vitro and in vivo [83,84,85]. Anticancer agents, including 5-fluorouracil (5-FU) and resibufogenin, caused necroptosis-mediated inhibition of human colorectal cancer cells in vitro and in vivo [86,87]. These results imply that the induction of necroptosis in vitro and in vivo may be an effective strategy for cancer therapy, at least according to the animal studies. However, clinical data based on the expression of necroptosis-related proteins in various types of cancers are characterized by variable associations with treatment prognosis. The expression of RIPK3 is decreased in patients with melanoma, breast cancer, and colorectal cancer, and high expression of RIPK3 in the tumors of these patients is strongly associated with survival [88,89,90]. In contrast to the clinical expression of RIPK3, high expression of RIPK1 in patients with lung cancer or pancreatic ductal adenocarcinoma is highly associated with poor prognosis [91,92]. Furthermore, reduced expression of MLKL has been reported to be associated with tumor metastasis into the lymph nodes [93]. Additionally, low expression of MLKL is strongly associated with poor prognosis in ovarian cancer patients [94]. Therefore, the correlation between necroptosis induction and its therapeutic effects in various types of cancer patients requires additional investigation.

### 3.2. Pyroptosis in Tumor Therapy

The major modulators of the pyroptosis process include the gasdermin family proteins [95]. GSDMA, GSDMB, GSDMC, GSDMD, GSDME (DFNA5), and DFNB59 constitute the gasdermin family and mediate pyroptosis in various types of cancer cells [96]. In particular, expression of GSDMA, GSDMB, GSDMC, or GSDME on the cell surface may lead to pore formation on the plasma membrane, e.g., in GSDMD-mediated pyroptosis [97]. The N-terminal domain of GSDMB was shown to form pores in an in vitro study; however, evidence is found unclear [98]. In general, GSDMA, GSDMC, GSDMD, and GSDME are known as tumor suppressors, and GSDMB was shown to function as an oncogene [99]. Since the differential expression of each gasdermin protein depends on cancer type, and remarkable data have been reported in the clinical studies, additional investigations are needed [100]. For example, enhanced expression of GSDMA and GSDMC was observed in human colorectal cancer progression [101]. Moreover, high expression of GSDMC in melanoma and lung adenocarcinoma was associated with a poor prognosis [102,103]. High levels of GSDMD are expressed in human NSCLC tissues, and knockdown of GSDMD in NSCLC cell lines (PC9, H1703, A549, SPC-A1, H1915, H1975, H1299 and H1650) attenuated their proliferation [104]. Elevated expression of GSDMD was observed in the tissues of patients with endometrioid carcinoma; however, other studies have shown that the tumor volume in groups treated with hydrogen-rich water was reduced by GSDMD-mediated pyroptosis compared to that in control mice [105]. High expression of GSDMB in cervical, uterine, esophageal, gastric, liver, and colon cancer tissues is significantly associated with a poor prognosis [98]. These results indicate that high expression of gasdermin proteins in the tumor tissues might not lead to favorable survival outcomes in cancer patients. Previous in vitro studies induced pyroptosis in cancer cells by treatment with various drugs. Metformin reduced cell viability by inducing pyroptosis caused by the cleavage of GSDMD in esophageal squamous cell carcinoma (ESCC) patients and human ESCC cell lines, including KYS510 and KYSE140 [106]. Nobiletin suppressed cell growth via GSDMD-mediated pyroptosis in ovarian cancer cells [107]. Navitoclax induced GSDME-related pyroptosis in human colorectal carcinoma HCT 116 cell [108]. In addition to the induction of pyroptosis by drug treatment, granzyme B secreted by cytotoxic T lymphocytes induced GSDME-mediated pyroptotic cell death without the formation of inflammasomes in mouse and human tumor cells [67]. Experimental data indicate that the induction of pyroptosis in the tumors may cause tumor suppression; however, additional clinical outcomes should be investigated in patients with various tumor types.

### 3.3. Ferroptosis in Tumor Therapy

Glutathione is an important molecule that regulates excessive ROS levels in the cells [109]. In particular, elevated activation of the KEAP1/NRF2/SLC7A11 axis of glutathione synthesis has been detected in various cancer cells and patients with breast cancer, bladder cancer, gastric carcinoma, glioma, and melanoma; this mechanism plays a major role in the resistance of cancer cells to cancer therapies [110,111]. Considering that ferroptosis is one of the types of RNCD induced by ROS-mediated lipid peroxidation process, i.e., the Fenton reaction, targeting this axis in cancer cells may be an alternative strategy for effective cancer treatment [112]. The induction of ferroptosis via inhibition of the glutathione synthesis pathway in various types of cancer cells has been investigated, and the results demonstrated that ferroptosis can effectively induce cell death in tumors resistant to radiotherapy and chemotherapy [34,113]. In previous studies, radiation alone induced lipid peroxidation and ferroptosis in a patient-derived xenograft (PDX) model [114]. In particular, iron chelation by desferrioxamine treatment of human glioma cells promoted the radioresistance of the cells in vitro [115]. These results indicate that ferroptosis may be an important strategy for the enhancement of radiosensitivity of tumors during radiotherapy. Ferroptosis inducers, such as erastin, sulfasalazine, and sorafenib, induced glutathione starvation and increased radiosensitivity in vitro and in vivo [116,117,118]. These ferroptosis inducers increase the sensitivity to anticancer drugs in various types of cancer during chemotherapy. For example, the induction of ferroptosis in cisplatin-resistant human cancer cells, such as head and neck cancer cells, non-small-cell lung cancer cells, and osteosarcoma cells, enhanced their sensitivity to cisplatin [119,120,121]. In clinical studies, high expression of SLC7A11 in patients with pulmonary cancer, pancreatic cancer, liver carcinoma, thyroid carcinoma, melanoma, or glioma was associated with a poor prognosis [122,123,124,125,126,127]. Therefore, the induction of ferroptosis may be an effective strategy for the treatment of therapy-resistant cancer types.

## 4. Conclusions

The induction of cell death in cancer cells is a major therapeutic strategy for tumor patients. Historically, the induction of apoptosis in various cancer cells has been tested in tumor therapies, and this strategy remains effective [128,129]. However, these approaches do not elicit a therapeutic effect in apoptosis-resistant tumors. To solve this problem, the induction of autophagic cell death has been attempted as a new strategy against apoptosis-resistant tumor cells [18,130]. The induction of autophagy in the cells can significantly reduce tumor volumes and may be a useful therapeutic strategy [131,132,133]. However, resistance to radiotherapy and chemotherapy can be induced by activation of autophagy, and autophagy has certain limitations in inducing cell death in various cancer types [134,135]. Therefore, the mechanisms of inflammatory cell death, such as RNCD, have been suggested for cancer treatment and were evaluated in various tumor studies.

RNCD has been studied in a variety of tumors. Necroptosis, pyroptosis, ferroptosis, parthanatos, and entotic cell death are the types of RNCD. Necroptosis, pyroptosis, and ferroptosis are discussed in this review. These types of cell death are caused by membrane disruption induced by specific signal cascades [136], which can elicit an inflammatory response within the tumor tissue [137]. The induction of RNCD in tumor tissues does not necessarily lead to positive results in cancer patients, since the inflammatory response in the tissues can have favorable or adverse effect depending on the tumor type [138]. However, significant antitumor effects of RNCD inducers were observed in anticancer drug-resistant cancer cells. FDA-approved homoharringtonine induced necroptosis in human colorectal and pancreatic cancers [139]. Simvastatin inhibited the proliferation and migration of human non-small-cell lung cancer through the induction of pyroptosis [140]. Moreover, simvastatin was administered to prevent the risk of recurrence in female patients after the breast cancer surgery [141]. Dihydrotanshinone, a component of Danshen (*Salvia miltiorrhiza* Bunge), was shown to induce ferroptosis in breast cancer cells and was reported to improve the survival of breast cancer patients [142]. Additionally, various RNCD inducers may inhibit tumor survival (Figure 3). Therefore, clinically meaningful results can be obtained in various additional studies.

To effectively induce RNCDs in cancer cells resistant to the apoptotic and autophagic pathways, the expression levels of mRNAs and proteins associated with RNCDs must be evaluated in patient tissues; then, the sensitivity of cancer cells to RNCDs must be enhanced by a combination of various therapeutic strategies; finally, the agents that directly activate RNCDs must be identified [143]. The results of the previous studies indicate that the expression levels of RNCD-related molecules in cancer patients differ depending on cancer types or individual genetic traits. The expression levels of MLKL were analyzed by immunohistochemistry in 613 patients with various cancers, including breast cancer, gastric cancer, cervical squamous cell carcinoma, ovarian cancer, colon cancer, and pancreatic adenocarcinoma; differences in the expression levels of MLKL were observed in patients with the same type of cancer, and low levels of MLKL expression were strongly associated with the advanced tumor stage and metastasis to the lymph nodes [93]. The expression level of GSDME was barely detectable or low in human breast cancer and melanoma cells, and low levels of GSDME in the cells were associated with resistance to anti-cancer therapies [144,145]. Moreover, in 89 gastric cancer patient tissues, gasdermin E gene was hypermethylated in approximately 50% of the samples, and 5-aza-2′deoxycytidine, a methyltrasferase inhibitor, suppressed the growth of primary gastric cancer cells [146]. Recently, cell–cell interactions have been reported to regulate ferroptosis. In human colorectal cancer cell lines, high density of the cells promoted the resistance to erastin, a ferroptosis inducer, via activation of the E-cadherin/Hippo pathway [147]. These results suggest that analysis of RNCD-related molecules in RNCD-resistant tumor tissues is important for effective anti-cancer therapy. The combination therapy for necroptosis induction has been extensively studied in acute myeloid leukemia (AML). The combination of emricasan, a stable and synergistic therapeutic effect in vitro and in vivo [148]. Cotreatment of ESCC with cisplatin and BI2536, a polo-like kinase 1 (PLK1) inhibitor, caused pyroptosis in vitro and in vivo [149]. Combined treatment of hypopharyngeal squamous carcinoma (HPSCC) cells with paclitaxel and RSL3 reduced the viability of the cells via ferroptosis [150]. Therefore, combination therapy adjusted to cancer properties may be an alternative therapeutic strategy for drug-resistant cancer types. Various drugs that directly induce RNCDs have been developed. HS-173, a potent PI3Kα inhibitor, induced necroptosis in human lung cancer cells in a RIP3-dependent manner [151]. A diazepin-quinazolin-amine derivative BIX-01294 enhanced GSDME-mediated pyroptosis in human gastric adenocarcinoma cell lines [152]. Talaroconvolutin isolated from the endophytic fungus *Talaromyces wortmanni* caused ferroptosis in human colorectal cancer in vitro and in vivo [153]. However, the number of clinical studies on the use of the induction of RNCDs in cancer patients is very limited, and additional investigation and clinical trials are required.

The induction of inflammatory responses in the tumor tissue may have beneficial or adverse effects on the host. Pro-inflammatory cytokines, including TNF-α, IL-1α, IL-1β, and IL-6, in the tumor microenvironment have been reported to promote tumor progression, metastasis, and invasion [154]. Additionally, the inflammatory responses caused by necroptosis, pyroptosis, and ferroptosis in the tumor cells can promote pro-tumor effects [101,155,156]. Therefore, there is no guarantee that the inflammatory response induced by RNCDs in the tumor microenvironment will be beneficial for the host. However, the antitumor immune response in the patient tumor tissues often depends on the immunogenicity of the tumor [157]. Induction of immunogenic cell death (ICD) alone is not sufficient to completely destroy the tumor microenvironment; however, improvement of immunogenicity in the tumor tissues is important for triggering dendritic cell-mediated anticancer immune responses [158]. Therefore, induction of ICD in the tumors may be an important strategy for anti-tumor therapy. Necroptosis, pyroptosis, and ferroptosis are the representative ICD mechanisms [137]. ICD in the tumor microenvironment plays an important role in the stimulation of the anti-tumor immune system. Immature dendritic cells that are chronically exposed to DAMPs released by ICD can mature and stimulate T cell responses that kill cancer cells via antigen presentation-dependent mechanisms [159]. Therefore, combination of RNCD inducers with appropriate immunotherapy may be an effective anticancer treatment strategy.

Beclin-1 is involved in the balance between apoptosis and autophagy [160]. The activity of caspase-8 is important for discrimination between apoptosis and necroptosis [161]. Caspase-3 activated in the process of apoptosis mediates pyroptosis [108]. Autophagy regulates ferroptosis through ferritin degradation [162]. These results indicate that the fate of cancer cells is not affected by only a single type of cell death. Therefore, tumor therapies can induce various types of cell death in tumor tissues, and cancer cells may activate resistance mechanisms to survive these challenges. Thus, effective suppression of cancer viability in tumor patients requires consideration of inherent characteristics of cancer to determine cell types that must be activated. In this review, the importance of RNCD for the induction of cell death of cancers resistant to apoptosis and autophagy inducers was discussed. Thus, although various trials investigating follow-up clinical outcomes of RNCD are needed, the use of RNCD inducers in patients with anticancer-drug resistant tumors may be a promising alternative therapeutic strategy.

## Figures and Tables

**Figure 1 cells-09-02709-f001:**
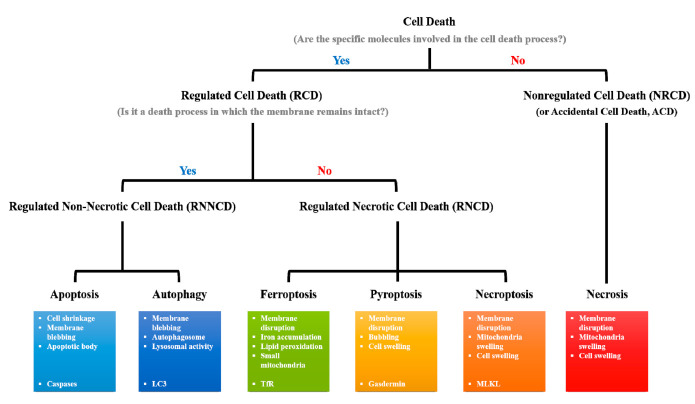
Classification of cell death types. Cell death types can be classified according to whether they are regulated by a specific signal pathway. Except for necrosis, most types of cell death are included in the RCD category. The type of RCD can also be classified according to membrane disruption. Apoptotic- and autophagic-cell death does not induce additional inflammatory responses, because their cellular components are sequestered within the membrane structures, such as apoptotic bodies and autophagosomes, respectively, during cell death processes. However, since RNCDs, including ferroptosis, pyroptosis, and necroptosis, are death types accompanied by membrane disruption, RNCDs can induce inflammatory responses in the tissues.

**Figure 2 cells-09-02709-f002:**
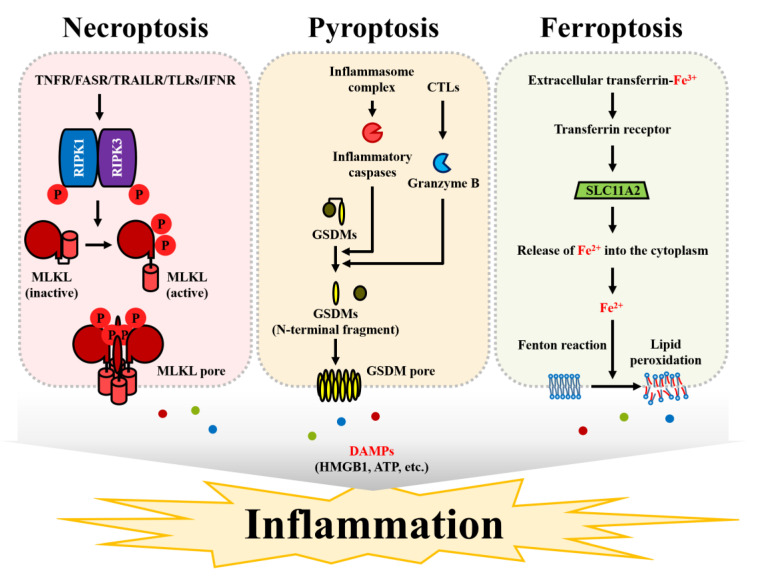
Types of RNCD. Necroptosis can be induced by the signals of TNF receptors, FAS receptors, TRAIL receptors, toll-like receptors, and interferon receptors. In the absence of caspase-8 activity, the phosphorylation of RIPK1 and RIPK3 induces phosphorylation and the subsequent activation of inactive MLKL. Activated MLKL is translocated into the plasma membrane and forms the pores approximately 4 nm in diameter. Pyroptosis is caused by gasdermin-mediated pore formation. The activation of the inflammasome complex induces the enzymatic activity of caspase-1, and caspase-1 cleaves gasdermin D. The N-terminal region of cleaved gasdermin D is translocated into the plasma membrane and forms the pores approximately 10–15 nm in diameter. In addition to caspase-1, gasdermin proteins can be cleaved by caspase-4, caspase-5, and caspase-11. Moreover, gasdermin E can be cleaved by granzyme B, which is released by cytotoxic CD8 T cells and natural killer (NK) cells. Ferroptosis is caused by membrane disruption induced by an iron-mediated lipid peroxidation process known as the Fenton reaction. Transferrin receptor transfers Fe^3+^ bound to extracellular transferrin into the intracellular endosomal regions. Fe^3+^ is reduced to Fe^2+^ by STEAP3, a ferrireductase, and generated Fe^2+^ is released into the cytoplasm by SLC11A2 (also known as divalent metal transporter 1, DMT1). The release of Fe^2+^ into the cytosol causes lipid peroxidation in the plasma membrane. As a result of RNCD induction, DAMPs, such as HMGB1 and ATP, can be released into the extracellular regions to induce inflammatory responses in the tissues.

**Figure 3 cells-09-02709-f003:**
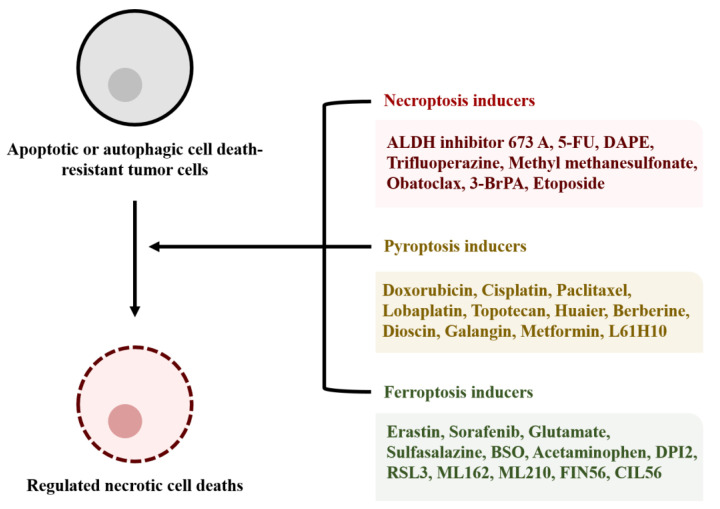
Inducers of RNCD. RNCD inducers can be used to induce cell death of tumor cells that are resistant to apoptotic or autophagic cell death. RNCD inducers can cause membrane disruption of the tumor cells and subsequent inflammatory responses in the tumor tissues. The appropriate administration of the inducers according to tumor types may be effective tumor therapies. Aldehyde dehydrogenase (ALDH1A) inhibitor; 5-fluorouracil (5-FU); 1,2-diarachidonoyl-sn-glycero-3-phosphoethanolamine (DAPE); 3-bromopyruvate (3-BrPA); buthionine sulfoximine (BSO); dihydroxyphenyl-imino-2-imidazolidine (DPI2); (1S,3R)-methyl 2-(2-chloroacetyl)-2,3,4,9-tetrahydro-1-[4-(methoxycarbonyl)phenyl]-1H-pyrido[3¨C-b]indole-3-carboxylate (RSL3); α-[(2-chloroacetyl)(3-chloro-4-methoxyphenyl)amino]-N-(2-phenylethyl)-2-thiopheneacetamide (ML162); [4-[bis(4-chlorophenyl)methyl]-1-piperazinyl](5-methyl-4-nitro-3-isoxazolyl)methanone (ML210); N2,N7-dicyclohexyl-9-(hydroxyimino)-9H-fluorene-2,7-disulfonamide (FIN56); 2,7-bis(1-piperidinylsulfonyl)-9H-fluoren-9-one, oxime (CIL56).

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
