# Peer review of "Regulated Necrotic Cell Death in Alternative Tumor Therapeutic Strategies"

_cells, 2020, doi:10.3390/cells9122709_

Round 1
Reviewer 1 Report
This is an interesting well-summarized review on regulated necrotic cell death (RND) in tumor cell death strategies.
The review is interesting but offers little depth. The contribution of this review to the field is very limited.
When citing/discussing the different studies, the authors need to distinguish between in vitro and in vivo studies to appreciate how well RND can be activated and contribute to cancer therapy.
Discuss what are the requirements for inducing RND in cancer cells.
The contribution of inflammation induced by RND to tumor progression must be discussed. It is also not clear from the review whether cancer cells can become resistant to RND.
Discuss in more depths the therapies that have been approved by the FDA and the ongoing clinical trials if any?
What are the challenges that lie ahead to exploit RND pathways in tumor therapeutic strategies.
May be focus the review on the most understood RND type?
Author Response
Response to Reviewers’ Comments
Reviewer 1
This is an interesting well-summarized review on regulated necrotic cell death (RND) in tumor cell death strategies.
The review is interesting but offers little depth. The contribution of this review to the field is very limited.
Q1. When citing/discussing the different studies, the authors need to distinguish between in vitro and in vivo studies to appreciate how well RND can be activated and contribute to cancer therapy.
Ans) We specifically indicate whether an experiment was performed in vivo or in vitro and with human cells or in an animal study. We have added the corresponding descriptions of each study in the revised manuscript.
Q2. Discuss what are the requirements for inducing RND in cancer cells.
Ans) We have added the requirements for the induction of RND in the revised manuscript. (9 pages, 335-367 lines)
Q3. The contribution of inflammation induced by RND to tumor progression must be discussed. It is also not clear from the review whether cancer cells can become resistant to RND.
Ans) We have added discussion of the contribution of the inflammation induced by RND in tumor progression (page 9, lines 346-362). Additionally, the description of cancer resistances to RNDs was added in the revised manuscript (page 9, lines 368-384).
Q4. Discuss in more depths the therapies that have been approved by the FDA and the ongoing clinical trials if any?
Ans) This is a legitimate question. We added a list of drugs approved by the FDA in the USA (page 8, lines 312-321).
Q5. What are the challenges that lie ahead to exploit RND pathways in tumor therapeutic strategies.
Ans) We have added the description of the anticipated challenges to the utilize of the RND pathways in tumor treatment strategies in the revised manuscript. (pages 9, lines 385-393)
Q6. May be focus the review on the most understood RND type?
Ans) This review is not focused on a specific RND type and describes all known types of RNDs, which can provide information about fundamental concepts to researchers and clinicians. Our team is working on ferroptosis in certain types of cancer, and we hope to present a review about ferroptosis in another publication.

Reviewer 2 Report
The manuscript titled “Regulated Necrotic Cell Death in Alternative Tumor 3 Therapeutic Strategies” by Woo et al., describes the need for induction of tumor cell death or Regulated cell deaths (RCDs) for effective treatment of cancers. The authors have provided detailed discussions on the mechanisms of various types of regulated necrotic cell death (RNCD), such as necroptosis, pyroptosis, and ferroptosis and their use in the treatment of various tumor types.
The only concern that was raised is: As depicted in Figure 1, the authors claim that TRAIL and TNF induce necrosis. While ample body of literature supports the fact that these two pathways are potent inducers of apoptosis, a more detailed comparison of apoptosis and necrosis induced by these two pathways and the basis for selectivity of these pathways to choose one pathway over the other (apoptosis vs necrosis) should be described in more details.
The manuscript is well written (requires some editorial assistance), clear and concise discussion of the data, conclusions are supported by the data presented, and the authors have used adequate number of relevant references.
Author Response
Response to Reviewers’ Comments
Reviewer 2
The manuscript titled “Regulated Necrotic Cell Death in Alternative Tumor 3 Therapeutic Strategies” by Woo et al., describes the need for induction of tumor cell death or Regulated cell deaths (RCDs) for effective treatment of cancers. The authors have provided detailed discussions on the mechanisms of various types of regulated necrotic cell death (RNCD), such as necroptosis, pyroptosis, and ferroptosis and their use in the treatment of various tumor types.
Q1. The only concern that was raised is: As depicted in Figure 1, the authors claim that TRAIL and TNF induce necrosis. While ample body of literature supports the fact that these two pathways are potent inducers of apoptosis, a more detailed comparison of apoptosis and necrosis induced by these two pathways and the basis for selectivity of these pathways to choose one pathway over the other (apoptosis vs necrosis) should be described in more details.
Ans) It is well known that TRAIL and TNF signaling induces apoptosis. However, under certain circumstances, cells can follow the necrotic pathway. Caspase 8 is an important molecule that can determinate between the apoptotic and necrotic cell death mechanisms. In the revised manuscript includes discussion of the conditions considered important for determination of each pathway. (page 4-5, lines 139-148)
The manuscript is well written (requires some editorial assistance), clear and concise discussion of the data, conclusions are supported by the data presented, and the authors have used adequate number of relevant references.

Reviewer 3 Report
This review is a well written comprehensive report on the different target pathways for regulated cell death and how these might be used for tumor therapy. The figures and graphs are instructive.
The English grammar needs some editing and I found a few typos.
Author Response
Response to Reviewers’ Comments
Reviewer 3
This review is a well written comprehensive report on the different target pathways for regulated cell death and how these might be used for tumor therapy. The figures and graphs are instructive.
Q1. The English grammar needs some editing and I found a few typos.
Ans) The revised manuscript was editing by a professional language-editing service. We upload the re-edited file.

Round 2
Reviewer 1 Report
I believe the authors should discuss more how RNCD patwhays can be used for cancer therapy and describe what are the challenges to achieve this goal.